**Data Availability Statement:** The institutional review board (IRB) of "Seoul Metropolitan Government-Seoul National University Boramae Medical Center" approved this retrospective study,

# Predictive factors affecting percutaneous drainage duration in the percutaneous treatment of common bile duct stones

**Min Uk Kim**[1], **Yoontaek Lee**[2]*, **Jae Hwan Lee**[3], **Soo Buem Cho**[4], **Myoung Seok Lee**[1], **Young Ho So**[1], **Young Ho Choi**[1]

**1** Department of Radiology, Seoul Metropolitan Government-Seoul National University Boramae Medical Center, Seoul, Korea, **2** Department of Surgery, Ewha Womans University Seoul Hospital, Seoul, Korea, **3** Department of Radiology, Seoul National University Bundang Hospital, Seongnam-si, Gyeonggi-do, Korea, **4** Department of Radiology, Ewha Womans University Seoul Hospital, Seoul, Korea

* noox01@gmail.com

## Abstract

The duration of percutaneous transhepatic biliary drainage (PTBD) is a critical factor that determines the duration of treatment. This study aimed to evaluate factors affecting the PTBD duration in patients who underwent percutaneous treatment of common bile duct (CBD) stones. This study analyzed data of 169 patients who underwent percutaneous treatment of CBD stones from June 2009 to June 2019. Demographic data, characteristics of stone, procedure-related factors, and laboratory findings before the insertion of PTBD tubes were retrospectively evaluated. To assess the effect of confounding factors on the PTBD duration, multivariate linear regression analysis was applied, incorporating significant predictive factors identified in the univariate regression analysis. In the univariate regression analysis, the predictive factor that showed high correlation with the PTBD duration was the initial total bilirubin level (coefficient = 0.68, $P < .001$) followed by the short diameter of the largest stone (coefficient = 0.19, $P = .056$), and previous endoscopic sphincterotomy (coefficient = -2.50, $P = .086$). The multivariate linear regression analysis showed that the initial total bilirubin level (coefficient = 0.50, $P < .001$) and short diameter of the largest stone (coefficient = 0.16, $P = .025$) were significantly related to the PTBD duration. The total bilirubin level before PTBD tube insertion and the short diameter of the largest CBD stone were predictive factors for the PTBD duration in patients who underwent percutaneous CBD stone removal. Careful assessment of these factors might help in predicting the treatment period, thereby improving the quality of patient care.

## Introduction

Common bile duct (CBD) stones are found in 11–21% of patients with gallstones and cause various clinical symptoms, such as biliary colic, cholangitis, acute pancreatitis, obstructive jaundice, and biliary sepsis [1, 2].

As a primary treatment of CBD stones, stone removal using endoscopic retrograde cholangiopancreatography (ERCP) has been favored since its inception in 1974 and is known to have

and the requirement for informed patient consent was waived. All data are anonymized and managed but include indirect identifiers (information sources) such as patients' age, sex, treatment dates, and treatment methods. The IRB of our institution does not allow disclosure of such data. The contact information of our IRB center is as follows: e-mail: brmirb02734@gmail.com Tel: +82-2-870-1851.

**Funding:** The author(s) received no specific funding for this work.

**Competing interests:** The authors have declared that no competing interests exist.

a success rate of up to 97% [3, 4]. However, endoscopic stone removal is difficult to access and likely to fail in cases of large stones, intrahepatic duct stone, biliary stricture, periampullary diverticulum, and history of gastrointestinal bypass surgery [5–9]. In these patients, initial treatment by percutaneous transhepatic biliary stone removal is known to be safe and effective [2, 8]. However, in the case of percutaneous CBD stone removal, the treatment period is longer than endoscopic treatment because percutaneous transhepatic biliary drainage (PTBD) is generally performed and the drainage tube is removed after several stone removal procedures.

PTBD duration is a critical factor that determines the duration of treatment and is an important factor that can have a significant influence on the quality of life of patients as well as on hospitalization and cost. If PTBD drainage duration can be predicted, it can be used to calculate treatment duration when starting treatment. Moreover, it can help with providing better information to the patients. However, to the best of our knowledge and considering the literature to date, no study has analyzed or described factors affecting the drainage period in the percutaneous treatment of CBD stones.

To fill this research gap, this study aimed to analyze factors affecting the drainage duration of PTBD when starting treatment in patients who have undergone percutaneous treatment of CBD stones.

## Materials and methods

### Study design and data collection

The institutional review board of Seoul Metropolitan Government-Seoul National University Boramae Medical Center approved this retrospective study (registration number; 10-2019-71), and the requirement for informed patient consent was waived. All procedures followed were in accordance with the ethical standards of the responsible committees on human experimentation (institutional and national) and with the Helsinki Declaration of 1964 and later versions. Medical records of consecutive adult hospitalized patients who underwent percutaneous treatment of CBD stones between June 2009 and June 2019 were collected from the electronic medical record database and picture archiving and communication system in our institution. Data such as laboratory blood tests, patient demographics, imaging findings, and treatment history were extracted from medical records. Among the patients whose CBD stone was identified through ERCP or CT images during this period, patients who received percutaneous treatment of CBD stone were finally included in the study. Patients with underlying hepatobiliary malignant tumors were excluded. The data were obtained and consolidated in November 2019.

### Percutaneous stone removal procedure

For all patients, PTBD was first performed to relieve clinical symptoms of cholangio-hepatitis and to create the approach route for the stone removal procedure. After injecting 5–10 mL of 2% lidocaine Hcl (Huons, Gyeonggi-do, Korea) into the skin, a 7 or 8.5-French drainage tube was inserted via puncture of the intrahepatic bile duct under ultrasonography and fluoroscopy guidance using a 21-gauge Chiba needle (Dukwoo Medical, Gyeonggi-do, Korea). Thereafter, the stone removal procedure was performed when the patient's clinical condition was stabilized a few days after inserting the drainage tube. Before removing the stone, cholangiography was performed to confirm the bile duct structure, stone size, and number of stones. Using the stone basket (Cook Medical, Bloomington, IN, USA), the stone was crushed, the remaining stone was captured directly, and the stone was removed percutaneously or pushed out to the duodenum. In some cases, it was also removed by pushing the stone to the duodenum using a balloon catheter. After the procedure, the PTBD catheter was reinserted, and follow-up cholangiography was performed several days later to confirm residual stone. If residual stone was

present, the above procedure was repeated to completely remove the stone. All CBD stones observed were removed, and if the contrast media passed smoothly into the duodenum on cholangiography, the drain catheter was removed.

### Factors associated with PTBD duration

The primary outcome of this study was the duration of PTBD, which was defined as the number of days from the date the drainage tube was first inserted until it was removed.

As regards the stone and procedure-related factors, we analyzed the tube size at the time of initial PTBD tube insertion, PTBD tube insertion site, previous endoscopic sphincterotomy (EST), number of stones, and stone size. Stone size was measured in the shortest diameter (d), longest diameter (D), and volume (V) of the largest stones on cholangiography. The volume was measured assuming that the stone was an ellipsoid, as shown below:

$$\left[ V = \frac{4\pi d^2 D}{3} \right]$$

As clinical factors, we analyzed age; sex; diabetes mellitus; peak levels of white blood cells (WBC), platelets, total bilirubin, amylase, creatine, prothrombin time-international normalized ratio (PT-INR), aspartate aminotransferase, alanine aminotransferase (ALT), alkaline phosphatase (ALP), γ-glutamyl transpeptidase (γ-GPT), C-reactive protein (CRP); and severity of cholangitis before PTBD tube insertion. The severity of cholangitis was classified as mild, moderate, and severe according to the Tokyo Guidelines for Acute Cholangitis [10].

### Statistical analysis

To assess the effect of confounding factors on the duration of PTBD, multivariate linear regression analysis with backwards stepwise elimination method was applied, incorporating significant predictive factors identified in the univariate regression analysis. Correlations were calculated using Pearson's correlation coefficient test. All variables with $P < .20$ in the univariate analysis were included in the multivariate analysis [11]. The coefficient and the standardized coefficient were indicated in order to clarify which of the independent variables measured in different units had a greater effect on PTBD duration.

All statistical analyses were performed with R software (http://cran.r-project.org/).

## Results

### Patient characteristics

Percutaneous CBD stone removal was performed on 173 patients who had unsuccessful or who declined endoscopic stone removal. Stones were successfully removed from 169 patients (technical success rate, 97.7%). Of the 173 patients, 107 (61.8%) attempted endoscopic approaches, but such approaches failed because of reasons such as periampullary diverticulum, previous gastrointestinal bypass surgery, or other anatomic difficulties. Of the 173 patients, 66 (38.2%) developed sepsis, pneumonia, and dementia, which hindered patients to cooperate and thus endoscopic removal. Of the 107 patients who attempted endoscopic stone removal, 32 underwent EST but not CBD cannulation. Finally, a total of 169 patients with successful percutaneous stone removal were included in this study.

The median duration of PTBD was 12 [interquartile range (IQR), 9–15; range, 3–48] days. Stone characteristics and procedure-related factors and clinical factors are summarized in Tables 1 and 2, respectively. The median short diameter of the largest stone was 11.7 mm

**Table 1. The characteristics of the stone and procedure-related factors.**

| | N = 169 |
|---|---|
| PTBD tube size | |
| 7fr | 10 (5.9%) |
| 8.5fr | 159 (94.1%) |
| PTBD site | |
| Lt. | 32 (18.9%) |
| Rt. | 137 (81.1%) |
| Previous EST | |
| no | 137 (81.1%) |
| yes | 32 (18.9%) |
| Number of stones | 1.0 [1.0;2.0] |
| Long diameter of largest stone (mm) | 16.1 [11.7;21.8] |
| Short diameter of largest stone (mm) | 11.7 [8.3;15.9] |
| Volume of largest stone (mm$^3$) | 8973.1 [3321.8;21387.8] |

Data shown are number (%), or median [IQR; interquartile range].

PTBD = percutaneous transhepatic biliary drainage; EST = endoscopic sphincterotomy.

(IQR, 8.3–15.9; range, 4.4–40.0), and the median level of total bilirubin was 2.6 mg/dL (IQR, 1.4–4.5; range, 0.4–15.5). Of the 169 patients, 20 (11.8%) had mild cholangitis, 107 (63.3%) had moderate cholangitis, and 42 (24.9%) had severe cholangitis.

**Table 2. The characteristics of clinical factors.**

| | N = 169 |
|---|---|
| Age (year) | 76.0 [68.0;82.0] |
| Sex | |
| Female | 72 (42.6%) |
| Male | 97 (57.4%) |
| Diabetes mellitus | |
| no | 130 (76.9%) |
| yes | 39 (23.1%) |
| Severity of cholangitis | |
| Mild | 20 (11.8%) |
| Moderated | 107 (63.3%) |
| Severe | 42 (24.9%) |
| Laboratory findings | |
| White blood cells count (x10$^3$/μℓ) | 10.3 [7.0;14.4] |
| Total bilirubin (mg/dL) | 2.6 [1.4;4.5] |
| Creatinine (mg/dL) | 0.8 [0.7;1.1] |
| Platelet (x10$^3$/μℓ) | 203.9 [152.0;245.0] |
| Prothrombin time (INR) | 1.1 [1.0;1.2] |
| Amylase (U/L) | 51.5 [33.0;89.0] |
| Aspartate aminotransferase (AST; IU/L) | 189.0 [70.0;410.0] |
| Alanine aminotransferase (ALT; IU/L) | 140.0 [48.0;242.0] |
| Alkaline phosphatase (ALP; IU/L) | 229.0 [140.0;350.0] |
| γ-glutamyl transpeptidase (γ-GPT; IU/L) | 262.0 [134.0;433.0] |
| C-reactive protein (mg/dL) | 5.4 [1.2;12.0] |

Data shown are number (%), or median [IQR; interquartile range].

**Table 3. Univariate regression analysis of predictive factors for PTBD duration.**

| | Coeff(B) | Std Coeff(β) | 95% CI of B | | P value |
| --- | --- | --- | --- | --- | --- |
| | | | Lower | Upper | |
| PTBD tube size (8.5fr) | -2.57 | -0.10 | -7.42 | 2.28 | 0.296 |
| PTBD site (Rt.) | 2.49 | 0.15 | -0.70 | 5.69 | 0.124 |
| Previous EST | -2.50 | -0.17 | -5.36 | 0.36 | 0.086 |
| Number of stones | 0.16 | 0.05 | -0.31 | 0.64 | 0.496 |
| Long diameter of largest stone | 0.07 | 0.10 | -0.06 | 0.20 | 0.321 |
| Short diameter of largest stone | 0.19 | 0.19 | 0.00 | 0.38 | 0.056 |
| Volume of largest stone | 0.00 | 0.10 | 0.00 | 0.00 | 0.314 |
| Age | 0.02 | 0.04 | -0.09 | 0.13 | 0.712 |
| Sex (Male) | -0.19 | -0.01 | -2.68 | 2.30 | 0.881 |
| Diabetes mellitus | -0.34 | -0.02 | -3.38 | 2.71 | 0.827 |
| Severity of cholangitis | 1.26 | 0.13 | -0.22 | 2.74 | 0.094 |
| White blood cells count(x1000) | -0.05 | -0.04 | -0.26 | 0.17 | 0.666 |
| Total bilirubin | 0.68 | 0.33 | 0.29 | 1.06 | 0.001 |
| Creatinine | 0.25 | 0.02 | -2.04 | 2.55 | 0.828 |
| Platelet (x1000) | 0.00 | -0.07 | -0.02 | 0.01 | 0.388 |
| Prothrombin time (INR) | 3.79 | 0.10 | -1.79 | 9.36 | 0.182 |
| Amylase | 0.00 | -0.02 | 0.00 | 0.00 | 0.824 |
| Aspartate aminotransferase | 0.00 | -0.04 | -0.01 | 0.00 | 0.724 |
| Alanine aminotransferase | 0.00 | 0.03 | 0.00 | 0.01 | 0.753 |
| Alkaline phosphatase | 0.00 | 0.13 | 0.00 | 0.01 | 0.198 |
| γ-glutamyl transpeptidase | 0.00 | 0.05 | 0.00 | 0.01 | 0.597 |
| C-reactive protein | 0.04 | 0.06 | -0.10 | 0.19 | 0.568 |

Coeff = coefficient; Std Coeff = standard coefficient; CI = confidence interval.

PTBD = percutaneous transhepatic biliary drainage; EST = endoscopic sphincterotomy.

## Factors associated with PTBD duration

The factors that were highly correlated with PTBD duration was initial total bilirubin level (coefficient = 0.68, $P < .001$), followed by the short diameter of the largest stone (coefficient = 0.19, $P = .056$), and previous EST (coefficient = -2.50, $P = .086$) on the univariate regression analysis (Table 3 and S1 Fig). The number of stones, initial PTBD tube size, PTBD tube insertion site, and peak levels of WBC, platelets, amylase, creatine, PT-INR, ALT, ALP, γ-GPT, and CRP showed no significant correlation with PTBD duration. In addition, the severity of cholangitis had no significant correlation with PTBD duration ($P = .094$).

In the multivariate regression analysis, the initial total bilirubin level (coefficient = 0.50, $P < .001$) and short diameter of the largest stone (coefficient = 0.16, $P = .025$) were significantly related to PTBD duration. The estimated multiple coefficient of determination ($R^2$) for the PTBD duration was 0.114 (Table 4).

## Discussion

In this retrospective study, we evaluated factors affecting the PTBD duration when starting treatment in patients who underwent percutaneous treatment of CBD stones.

Percutaneous transhepatic biliary stone removal is currently considered the second-line therapy after a failed ERCP. Percutaneous biliary stone removal through the PTBD route was first reported in 1962 by Monte et al. [12], and it became practical with the use of stone basket

**Table 4. Multivariate regression analysis of predictive factors for PTBD duration.**

| | Coeff(B) | Std Coeff(β) | 95% CI of B | | P value |
| --- | --- | --- | --- | --- | --- |
| | | | Lower | Upper | |
| Previous EST | -2.08 | -0.14 | -4.22 | 0.07 | 0.057 |
| Short diameter of largest stone | 0.16 | 0.17 | 0.02 | 0.30 | 0.025 |
| Total bilirubin | 0.50 | 0.24 | 0.20 | 0.81 | <0.001 |

Linear regression analysis using backward stepwise elimination.

Coeff = coefficient; Std Coeff = standard coefficient; CI = confidence interval.

coefficient of determination (R2) = 0.114; P < 0.001.

EST = endoscopic sphincterotomy.

catheters by Burhenne et al. [13, 14]. Recently, it is also used to approach the PTBD route and to push the stones out of the duodenum using a balloon catheter [15]. Compared with endoscopic stone removal, percutaneous biliary stone removal has a shorter approach pathway that makes it easier to maintain the direction in which the stones are pushed to the duodenum and the direction of forces applied. Therefore, we can increase the intensity of the force to remove the stone. In our study, the success rate of percutaneous CBD stone removal treatment was 97.7%. Previous studies reported that the success rates of endoscopic CBD stone removal ranged from 90% to 97% and that of percutaneous CBD stone removal ranged from 87% to 94% [3, 4, 16].

In this study, the median duration of PTBD was 12 (range, 3–48) days. Van der Velden et al. reported 16 (range, 3–299) days as the median time between percutaneous drainage and last cholangiography in patients who underwent percutaneous treatment of bile duct stones [16].

The standard endoscopic CBD stone removal procedure is to first perform EST and then to remove the CBD stone using a balloon catheter or stone basket. In some cases, EST is possible, but CBD stone removal is not possible because of failed CBD cannulation due to anatomical difficulties. In our study, 32 of 107 patients (29.9%) who attempted endoscopic removal underwent EST, but CBD cannulation was unsuccessful. Biliary decompression with EST is beneficial in patients with biliary obstruction due to CBD stones. Moreover, Hui et al. reported that EST performed during biliary drainage can decrease the duration of symptoms and hospital stay in patients with acute cholangitis without CBD stones [17]. EST is thought to improve the internal drainage of the bile and facilitate discharge of CBD stones. For comparison, the PTBD duration of the EST groups was shorter than that of the non-EST group, but it was not statistically significant in our multivariate regression analysis (coefficient = -2.08, $P$ = .057).

In our univariate regression analysis, the short diameter of the largest stone was more affected by the PTBD duration than the long diameter or volume of the largest stone. In the multivariate regression analysis of stone and procedure-related factors in our study, the short diameter of the largest stone was the only factor significantly correlated with the PTBD duration ($P$ = .025). Generally, bile duct stones up to 1.5 cm in diameter can be extracted intact after EST, and the rate of successful extraction decrease with increasing size of the stone. If the stone is large, it can be impacted in the CBD, difficult to remove, and can worsen obstructive cholangitis [18–20]. The size of the CBD stone is usually measured by the long diameter of the largest stone. However, the actual CBD stone has an ellipsoid form. Ellipsoid CBD stones pass by the longitudinal axis as they pass through the papilla or percutaneous route. Hence, even if the longitudinal long axis is large, the CBD stone can easily pass through the papilla or percutaneous route if the transverse short axis is small. Conversely, the larger the size of the short axis,

the more likely it is to be impacted in the CBD and papilla. Therefore, the larger the short diameter of the CBD stone, the more difficult it can be removed and thus worsen the obstruction.

CBD stones cause acute and recurrent cholangitis. Partial or complete biliary obstruction and subsequent infection is the major factor in the development of acute cholangitis. Various clinal factors are known as predictive factors of acute cholangitis. According to the Tokyo Guidelines for Acute Cholangitis, hypotension, disturbance of consciousness, respiratory dysfunction, creatinine, PT-INR, WBC, platelet, fever, age, hyperbilirubinemia, and hypoalbuminemia are assessment criteria for the severity of cholangitis [10]. In our study, the severity of cholangitis had no significant correlation with PTBD duration ($P$ = .094). Of the many clinical factors in our study, the total bilirubin level was the only factor significantly correlated with the PTBD duration. The total bilirubin level has high specificity and positive and negative predictive values among other biochemical laboratory measurements in the diagnosis of acute biliary obstruction [21]. In addition, the total bilirubin level is a useful predictor of persistent CBD stone in gallstone pancreatitis, and it is used as a factor in the mortality prediction of acute suppurative cholangitis [22, 23]. The total bilirubin level well reflects the degree of biliary obstruction; we consider that normalization of the total bilirubin level rather than other clinical factors is the main factor affecting the PTBD duration.

This study has some limitations. The retrospective and single-institution design may lead to patient selection bias. In addition, we analyzed patients who declined or had failed endoscopic treatment. In the future, large-scale and multicenter prospective studies are warranted.

In conclusion, the total bilirubin level before the PTBD tube insertion and the short diameter of the largest CBD stone were predictive factors for the duration of PTBD in patients who underwent percutaneous CBD stone removal. Careful assessment of these factors might help the prediction of the treatment period, thereby improving the quality of patient care.

## Supporting information

**S1 Fig. The relation between the correlated factors and the duration of PTBD.** A, previous EST (coefficient = -2.50, P = .086). B, short diameter of the largest stone (coefficient = 0.19, $P$ = .056). C, total bilirubin level (coefficient = 0.68, $P < $ .001).
(TIF)

## Acknowledgments

We would like to thank Editage (www.editage.co.kr) for English language editing.

## Author Contributions

**Formal analysis:** Min Uk Kim, Yoontaek Lee.

**Methodology:** Soo Buem Cho.

**Supervision:** Young Ho So, Young Ho Choi.

**Writing – original draft:** Min Uk Kim, Yoontaek Lee.

**Writing – review & editing:** Min Uk Kim, Yoontaek Lee, Jae Hwan Lee, Soo Buem Cho, Myoung Seok Lee, Young Ho So, Young Ho Choi.

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
