## [Decision Letter · Decision Letter 0]

4 Dec 2020

PONE-D-20-31996

Predictive factors affecting percutaneous drainage duration in the percutaneous treatment of common bile duct stones

PLOS ONE

Dear Dr. Lee,

Thank you for submitting your manuscript to PLOS ONE. After careful consideration, we feel that it has merit but does not fully meet PLOS ONE’s publication criteria as it currently stands. Therefore, we invite you to submit a revised version of the manuscript that addresses the points raised during the review process.

First of all, thank you for submitting PLOS ONE journal. This report is very well written and please respond our comments.

We look forward to receiving your revised manuscript.

Kind regards,

Yutaka Kondo

Academic Editor

PLOS ONE

Additional Editor Comments:

This manuscript is well written although the methods is unclear.

1. Why the authors used all variables with P < .20 ? Please explain the rationale with citation.

2. The authors reported Coeff(B) and Std Coeff(β) in table 3and 4. Why the authors reported both? What is the differences these two methods?

In addition, the authors did not mention the Coeff(B) and Std Coeff(β) in method. Please explain in the methods.

Journal Requirements:

2. Thank you for stating in the text of your manuscript "the requirement for informed patient consent was waived". Please also add this information to your ethics statement in the online submission form.

3. Please provide more information on data extraction and collection. Please provide:

a) the date the data was obtained by researchers

b) any inclusion criteria used to include patients (we note that you already state exclusion criteria)

c) a list of the data extracted from medical records.

6. Please include your tables as part of your main manuscript and remove the individual files. Please note that supplementary tables should be uploaded as separate "supporting information" files.

Reviewers' comments:

Reviewer's Responses to Questions

**Comments to the Author**

1. Is the manuscript technically sound, and do the data support the conclusions?

Reviewer #1: Yes

Reviewer #2: Yes

2. Has the statistical analysis been performed appropriately and rigorously? 

Reviewer #1: Yes

Reviewer #2: No

3. Have the authors made all data underlying the findings in their manuscript fully available?

Reviewer #1: No

Reviewer #2: Yes

4. Is the manuscript presented in an intelligible fashion and written in standard English?

Reviewer #1: Yes

Reviewer #2: Yes

5. Review Comments to the Author

Reviewer #1: Overall, the idea to analyze factors affecting PTBD treatment duration is novel, the manuscript also well written and very concise. There are few questions author should address:

1/ What is the reason to choose P < .20 for including variables in univariate regression analysis into multivariate regression analysis for the context of this research topic? Is there any reference for it? Please provide

2/ Why despite variable EST not significant in the univariate analysis was included into multivariate analysis? Is there any theoretical base for it? Please provide

Reviewer #2: Congratulations on a well written paper on a less addressed topic.

The research question needs to be mentioned with clarity in the introduction.

Was a correlation analysis done between the continuous variables and duration of drainage by scatter plot? This is not clear from your methodology section. You need to describe in detail how the univariate analysis was arrived at.

What is the take home message from the study based on your findings? Currently you mention it as a factor to decide the choice of treatment; however you also mention that percutaneous drainage is alrady a second line option. SO what is the implication of your paper?

6. PLOS authors have the option to publish the peer review history of their article (what does this mean?). If published, this will include your full peer review and any attached files.

Reviewer #1: No

Reviewer #2: **Yes: **Aneesh Basheer

---

## [Author Response · Author response to Decision Letter 0]

28 Dec 2020

Reviewer #1: 

1. What is the reason to choose P < .20 for including variables in univariate regression analysis into multivariate regression analysis for the context of this research topic? Is there any reference for it? Please provide

Model minimization was performed by stepwise backward elimination method. To rule out confounding more effectively, we chose P < 0.2 to identify as many potential predictors and confounders as possible at the cost of multiple testing. (Maldonado G, Greenland S. Simulation study of confounder-selection strategies. Am J Epidemiol. 1993;138:923-936. DOI: 10.1093/oxfordjournals.aje.a116813.)

We revised and added the phrase as follows, in “Methods”

“…multivariate linear regression analysis with backwards stepwise elimination method was applied, incorporating significant predictive factors identified in the univariate regression analysis”

2. Why despite variable EST not significant in the univariate analysis was included into multivariate analysis? Is there any theoretical base for it? Please provide

As described above, P < 0.2 was accepted as significant in the selection of the risk factors.

Reviewer #2: 

1. Was a correlation analysis done between the continuous variables and duration of drainage by scatter plot? This is not clear from your methodology section. You need to describe in detail how the univariate analysis was arrived at.

We used linear regression analysis to examine a correlation between the continuous variables and duration of drainage (Pearson’s correlation coefficient test). We added scatter plots of correlated factors (initial total bilirubin level, the short diameter of the largest stone, previous EST) as supplementary tables and the phrase as follows, in “Methods”

“Correlations were calculated using Pearson’s correlation coefficient test.”

2. What is the take home message from the study based on your findings? Currently you mention it as a factor to decide the choice of treatment; however you also mention that percutaneous drainage is alrady a second line option. SO what is the implication of your paper?

The “treatment plan” mentioned in the conclusion implies not only to the treatment of CBD stones, but also to the overall management of patient care (such as management of associated underlying diseases). 

To clarify the above points, we revised and added the phrase as follows, as the last sentence of “discussion” and conclusion of “abstract”

“Careful assessment of these factors might help the prediction of the treatment period, thereby improving the quality of patient care.”

---

## [Decision Letter · Decision Letter 1]

18 Feb 2021

Predictive factors affecting percutaneous drainage duration in the percutaneous treatment of common bile duct stones

PONE-D-20-31996R1

Dear Dr. Lee,

We’re pleased to inform you that your manuscript has been judged scientifically suitable for publication and will be formally accepted for publication once it meets all outstanding technical requirements.

Kind regards,

Yutaka Kondo

Academic Editor

PLOS ONE

Additional Editor Comments (optional):

Reviewers' comments:

Reviewer's Responses to Questions

**Comments to the Author**

1. If the authors have adequately addressed your comments raised in a previous round of review and you feel that this manuscript is now acceptable for publication, you may indicate that here to bypass the “Comments to the Author” section, enter your conflict of interest statement in the “Confidential to Editor” section, and submit your "Accept" recommendation.

Reviewer #2: All comments have been addressed

2. Is the manuscript technically sound, and do the data support the conclusions?

Reviewer #2: Yes

3. Has the statistical analysis been performed appropriately and rigorously? 

Reviewer #2: Yes

4. Have the authors made all data underlying the findings in their manuscript fully available?

Reviewer #2: Yes

5. Is the manuscript presented in an intelligible fashion and written in standard English?

Reviewer #2: Yes

6. Review Comments to the Author

Reviewer #2: You have addressed the concerns raised in the review appropriately and adequately. The manuscript addresses an important area in medical care.

7. PLOS authors have the option to publish the peer review history of their article (what does this mean?). If published, this will include your full peer review and any attached files.

Reviewer #2: **Yes: **Aneesh Basheer

---

## [Editor Report · Acceptance letter]

22 Feb 2021

PONE-D-20-31996R1 

Predictive factors affecting percutaneous drainage duration in the percutaneous treatment of common bile duct stones 

Dear Dr. Lee:

I'm pleased to inform you that your manuscript has been deemed suitable for publication in PLOS ONE. Congratulations! Your manuscript is now with our production department. 

Kind regards, 

on behalf of

Dr. Yutaka Kondo 

Academic Editor

PLOS ONE